# Infliximab Efficacy May Be Linked to Full TNF-α Blockade in Peripheral Compartment—A Double Central-Peripheral Target-Mediated Drug Disposition (TMDD) Model

**DOI:** 10.3390/pharmaceutics13111821

**Published:** 2021-11-01

**Authors:** David Ternant, Olivier Le Tilly, Laurence Picon, Driffa Moussata, Christophe Passot, Theodora Bejan-Angoulvant, Céline Desvignes, Denis Mulleman, Philippe Goupille, Gilles Paintaud

**Affiliations:** 1EA 4245 “Transplantation, Immunology, Inflammation”, Université de Tours, 37044 Tours, France; letilly@univ-tours.fr (O.L.T.); driffa.moussata@univ-tours.fr (D.M.); theodora.angoulvant@univ-tours.fr (T.B.-A.); celine.desvignes@univ-tours.fr (C.D.); gilles.paintaud@univ-tours.fr (G.P.); 2Department of Clinical Pharmacology, CHRU de Tours, 37044 Tour, France; 3Department of Gastroenterology, CHRU de Tours, 37044 Tour, France; L.PICON@chu-tours.fr; 4Département de Biopathologie, Institut de Cancérologie de l’Ouest, 49055 Angers, France; Christophe.Passot@ico.unicancer.fr; 5EA 7501 “Groupe Innovation et Ciblage Cellulaire”, Université de Tours, 37044 Tour, France; denis.mulleman@univ-tours.fr; 6Department of Rheumatology, CHRU de Tours, 37044 Tour, France; philippe.goupille@univ-tours.fr

**Keywords:** infliximab, pharmacokinetics, target-mediated drug disposition, inflammatory bowel diseases

## Abstract

Infliximab is an anti-TNF-α monoclonal antibody approved in chronic inflammatory bowel diseases (IBD). This study aimed at providing an in-depth description of infliximab target-mediated pharmacokinetics in 133 IBD patients treated with 5 mg/kg infliximab at weeks 0, 2, 14, and 22. A two-compartment model with double target-mediated drug disposition (TMDD) in both central and peripheral compartments was developed, using a rich database of 26 ankylosing spondylitis patients as a reference for linear elimination kinetics. Population approach and quasi-steady-state (QSS) approximation were used. Concentration-time data were satisfactorily described using the double-TMDD model. Target-mediated parameters of central and peripheral compartments were respectively baseline TNF concentrations (R^C^_0_ = 3.3 nM and R^P^_0_ = 0.46 nM), steady-stated dissociation rates (K^C^_SS_ = 15.4 nM and K^P^_SS_ = 0.49 nM), and first-order elimination rates of complexes (k^C^_int_ = 0.17 day^−1^ and k^P^_int_ = 0.0079 day^−1^). This model showed slower turnover of targets and infliximab-TNF complex elimination rate in peripheral compartment than in central compartment. This study allowed a better understanding of the multi-scale target-mediated pharmacokinetics of infliximab. This model could be useful to improve model-based therapeutic drug monitoring of infliximab in IBD patients.

## 1. Introduction

Infliximab is a chimeric monoclonal antibody (mAb) binding to tumor necrosis factor alpha (TNF-α), approved in chronic inflammatory diseases both in rheumatology—e.g., rheumatoid arthritis (RA) and ankylosing spondylitis (AS)—and in inflammatory bowel diseases (IBD)—e.g., Crohn’s disease (CD) or ulcerative colitis (UC). The administration of weight-adjusted infliximab doses leads to highly variable infliximab concentrations. This variability is relevant since infliximab concentrations were shown to be associated with clinical efficacy, especially in IBD [1]. Infliximab pharmacokinetics was analyzed using compartmental modeling in more than 30 studies to date. Some of these studies reported predictive models suitable for model-based therapeutic drug monitoring of infliximab [2,3,4]. The aim of individual dosing regimen to maintain serum trough concentrations of infliximab above target concentrations associated with good clinical response. In IBD, these target concentrations were reported to be 3–5 mg/L at steady-state [3].

Infliximab binds to TNF-α with high affinity [5,6,7]. This leads to the formation of infliximab-TNF complexes that are cleared by the immune system and to a mutual consumption of infliximab and TNF-α. Therefore, to allow a good clinical response, infliximab should be in stoichiometric excess compared to TNF-α during all over the time of treatment. This is the case for blood TNF-α, since infliximab concentrations are more than 10,000-fold higher than TNF-α levels [8,9,10]. However, a recent study from our group suggested that antigen mass (i.e., total amount of TNF-α able to interact with infliximab) is more than 200 fold higher than circulating TNF-α and that trough infliximab serum concentrations above target values do not lead to sustained TNF-α inhibition [11]. In IBD, since this phenomenon is not associated with systematic loss of response, and that TNF-α reservoir is admitted to be both circulating and expressed on intestine inflammatory cells (monocytes, macrophages) [12,13,14], it may be hypothesized that infliximab’s effect is related to its binding to TNF-α in a ‘deep’ compartment.

The joint kinetics of both infliximab and target may be described using target-mediated drug disposition (TMDD) models [15,16]. Up to date, five publications reported target-mediated pharmacokinetics of infliximab in IBD [11,17,18,19,20]. However, none of them allowed the description of the interactions of infliximab and TNF-α in the deep compartment. Indeed, four of these studies used one-compartment pharmacokinetic models [11,18,19,20], while one used a two-compartment TMDD model which described interactions between infliximab and circulating TNF-α levels in the central compartment [17].

Therefore, the present study aimed at describing infliximab target-mediated pharmacokinetics in IBD patients, allowing the quantification of infliximab interactions with TNF-α in both central and peripheral compartments using population TMDD modeling.

## 2. Methods

### 2.1. Data

The present study was conducted using two cohorts of patients:-A bicentric study of 26 ankylosing spondylitis (AS) patients (SPAXIM, NCT00607403). Inclusion and exclusion criteria were previously described [21]. Patients were treated with infliximab 5 mg/kg infusions at weeks 0, 2, 6, 12, and 18; blood samples were collected to measure infliximab concentrations before, 2 and 4 h after each infusion, and at each intermediate visit at weeks 1, 3, 4, 5, 8, 10, and 14 [21]. One patient was not assessed in the present work because he developed anti-drug antibodies (ADA) starting from the second infliximab administration;-A retrospective cohort of 133 routine IBD patients treated with infliximab between 2006 and 2012 in the Tours University Hospital (Tours, France). For these patients, individual results were interpreted, sent to the prescriber and discussed in clinic-biological rounds. Infliximab concentrations were therefore not sought for this study and were already used in previous publications [22,23]. This cohort included patients with trough and peak infliximab concentrations measured during treatment initiation and in whom anti-drug antibodies (ADA) were not detected at least within the three first infliximab cycles. Patients were excluded if less than three concentration values were available (which excluded seven patients of our database between 2006 and 2012), if no peak concentrations were available (which excluded two more patients) and if ADA were detected at first, second of third cycles (which excluded three more patients). Of note, if ADA were detected starting for the fourth cycle, patients were not excluded from analysis, but data regarding cycles with ADA were discarded (eight patients).

Infliximab concentrations were measured using a validated enzyme-linked immunosorbent assay (ELISA). Detection, and lower and upper quantitation limits were 0.031, 0.103, and 15 mg/L (), respectively. This assay was shown to measure unbound infliximab concentrations [24]

### 2.2. Model Development

#### 2.2.1. Structural Model Design

A double TMDD model with quasi-steady-state (QSS) approximation [25,26] accounting for interactions of infliximab with TNF in both central and peripheral compartments in IBD patients was developed (Figure 1, Appendix B). As in our previous study, target-mediated pharmacokinetic parameters of IBD were obtained using AS as a reference of linear pharmacokinetics. Indeed, no target influence was detected in AS despite a dense sampling protocol [27]. Levels of TNF-α were not measured and considered as latent variable. The double QSS model was
$$\frac{dC_{T}}{dt}=In\left(t\right)-\frac{CL}{V_{1}}.C-\frac{Q}{V_{1}}.C+\frac{Q}{V_{2}}.C_{P}-k_{int}^{C}.\left(C_{T}-C\right)$$
$$\frac{dR_{T}^{C}}{dt}=k_{in}^{C}-k_{out}.\left(R_{T}^{\;C}-C_{T}+C\right)-k_{int}^{C}.\left(C_{T}-C\right)$$
$$C=\frac{1}{2}\left[\left(C_{T}-R_{T}^{C}-K_{SS}^{C}\right)+\sqrt{{\left(C_{T}-R_{T}^{C}-K_{SS}^{C}\right)}^{2}-4.K_{SS}^{C}.C_{T}}\right]$$
$$\frac{dC_{PT}}{dt}=\frac{Q}{V_{1}}.C-\frac{Q}{V_{2}}.C_{P}-k_{int}^{P}.\left(C_{PT}-C_{P}\right)$$
$$\frac{dR_{T}^{P}}{dt}=k_{in}^{P}-k_{out}.\left(R_{T}^{\;P}-C_{PT}+C_{P}\right)-k_{int}^{P}.\left(C_{PT}-C_{P}\right)$$
$$C_{P}=\frac{1}{2}\left[\left(C_{PT}-R_{T}^{P}-K_{SS}^{P}\right)+\sqrt{{\left(C_{PT}-R_{T}^{P}-K_{SS}^{P}\right)}^{2}-4.K_{SS}^{P}.C_{PT}}\right]$$
where *In(t)* is infliximab input function; *C* and *C_T_* are unbound and total infliximab concentrations in central compartment, respectively; *C_P_* and *C_PT_* are unbound and total infliximab concentrations in peripheral compartment, respectively; *V*_1_ and *V*_2_ are central and peripheral volumes of distribution, respectively; *CL* and *Q* are systemic and intercompartmental clearances, respectively; $R_{T}^{C}$ and $R_{T}^{P}$ are latent total TNF-α levels interacting with infliximab in central and peripheral compartments, respectively; $k_{in}^{C}$ and $k_{in}^{P}$ are zero-order TNF-α input relative to central and peripheral compartments, respectively; $K_{SS}^{C}$ and $K_{SS}^{P}$ are steady-state dissociation constants relative to central and peripheral compartments, respectively; $k_{int}^{C}$ and $k_{int}^{P}$ are first-order elimination rate constants of infliximab-TNF complexes; and $k_{out}$ is first-order TNF-α elimination rate constant. Rather than $k_{in}^{C}$ and $k_{in}^{P}$, baseline levels in central ($R_{0}^{C}=k_{in}^{C}/k_{out}$) and peripheral ($R_{0}^{P}=k_{in}^{P}/k_{out}$) compartments were estimated. Pharmacokinetic data were analyzed using the nonlinear mixed-effect modeling software MONOLIX Suite 2020 (Lixoft^®^, Antony, France). A large number of iterations (100 and 400 iteration kernels 1 and 2, respectively) and five Markov chains were used. Objective function (−2LL) and Fisher information matrix were computed using importance sampling and stochastic approximation methods, respectively. All parameters were estimated simultaneously. The structural target-mediated pharmacokinetic model of infliximab was developed in four steps:1.Bicompartimental: two-compartment model with no target interaction;2.Central TMDD: two-compartment model with target interaction in central compartment;3.Peripheral TMDD: two-compartment model with target interaction in peripheral compartment;4.Double TMDD: two-compartment model with target interaction in both central and peripheral compartments.

Early attempts showed that some parameters could not be estimated and had to be fixed; best strategy was to fix the value of $k_{out}$. Several values were derived from serum half-life estimations in humans (40.6 day^−1^ [28]), rhesus monkeys (10 or 66 day^−1^ [29]), rats (33 or 154 day^−1^ [30]), and rabbits (166 day^−1^ [31]). Based on these possible values, the double QSS model was run while fixing k_out_ at the following values: 5, 10, 20, 40, 100, 150, or 200 day^−1^. The value leading to best model performances was retained. This was done on the central TMDD model and verified on the double TMDD model while fixing the same k_out_ value for both central and peripheral compartments.

#### 2.2.2. Statistical Models

Interindividual and error models

Statistical model for interindividual variability was exponential, where interindividual variances were fixed to 0 if relative standard errors (RSE) and/or shrinkage values were high. Error model was mixed additive-proportional as in our previous analyses of these data [21,22].

Influence of covariates

Covariates that were considered in this analysis were those which were found significantly associated with pharmacokinetic parameter in previous studies involving these databases—i.e., body weight (BW), sex (SX), and underlying disease [3,21,22,32]. The continuous covariate BW coded as a power function and centered on its median, whereas categorical covariates (CAT) were SX and underlying IBD disease (DIS = CD or UC); IBD was tested only on IBD patients. The influence of CAT was implemented as: $\ln\;\left(\theta_{\mathrm{TV}}\right)=\ln\;\left(\theta_{\mathrm{CAT}=0}\right)+{\;\beta }_{\mathrm{CAT}=1}$, where $\theta_{\mathrm{TV}}$ is typical value of structural parameter θ, θ_CAT=0_ is the value of θ for the reference category, and β_CAT=1_ is a parameter leading to the value for the other category. References for SX and DIS were females, and CD, respectively. The association of covariates with parameters was tested only with parameters for which interindividual variances was estimable.

#### 2.2.3. Model Evaluation

Model comparison

Structural and statistical models were compared using objective functions. The best structural model was the one with the lowest Akaike’s information criterion (AIC). This criterion combines the −2.ln likelihood (−2LL) and twice the number of parameters to be estimated. Statistical (interindividual, convariate) models were compared using the likelihood ratio test (LRT), where the difference in −2LL between nested models was assumed to follow a χ^2^ distribution. The association of covariate with parameter distribution was assessed in two steps. First, a univariate step where the association of each covariate on parameter was tested separately. Covariates significantly associated with parameters (α < 0.05) were added in the full model. Second, a multivariate step was made, where covariates of the full model were individually removed from the full model (backward stepwise procedure). Covariates were kept in the final model if their removal resulted in a significant increase in −2LL (α < 0.02).

Model goodness of fit

Models were evaluated graphically using goodness-of-fit diagnostic plots: observed vs. population (PRED) and individual (IPRED) fitted concentrations; population (WRES) and individual (IWRES) weighted residuals vs. PRED and IPRED, respectively. Visual predictive checks and normalized prediction distribution errors (NPDE) were performed by simulating 1000 replicates using both fixed and random parameters of the final model.

### 2.3. Simulations

The objective of simulations was to evaluate TNF-α blockade, which was quantified using total ($R_{T}^{C}$ and $R_{T}^{P}$), unbound ($R^{C}$ and $R^{P}$) target levels, as well as ratios of unbound/total target in central ($R^{C}/R_{T}^{C}$) and peripheral ($R^{P}/R_{T}^{P}$) compartments, respectively. Structural and interindividual parameter values of the final model were used to simulate median and 90% prediction intervals of infliximab concentrations and target kinetics in both compartments. Simulated dosing regimen was 5 mg/kg infused at weeks 0, 2, 6, 14, and 22. Ratios of 50/50 and 5/1 for male/female and CD/UC were assumed. Distribution of BW was assumed as Gaussian with mean and standard deviation of 66 kg and 15 kg, respectively, with values restricted within the population range (41–110). Simulations were made using Simulx 2020 (Lixoft^®^, Antony, France).

## 3. Results

### 3.1. Base Model

A total of 1333 infliximab concentrations were available in the assessed 158 patients (Table 1, Appendix A). The double TMDD model allowed the best description of concentration-time data since it led to the deepest decrease in AIC (Table 2). Optimal fixed value for *k_out_* was 20 day^−1^ (Appendix A). All structural, interindividual and residual parameters of base and final model were estimated with good accuracy (Table 2), including V_1_, V_2_, CL, $R_{0}^{C}$, and $R_{0}^{P}$. Interindividual variances of other parameters were poorly identifiable and were set to 0. Diagnostic plots (Figure 2) showed a good agreement between observed and model-fitted concentrations. Residuals, VPCs and NPDEs displayed neither bias nor model misspecification (Figure 2). Endogenous (i.e., non-target-mediated) elimination half-life (T½-β) was approximately 17 days.

### 3.2. Final Model

During the univariate step, BW and SX significantly influenced both V_1_ and CL, while IBD significantly influenced $R_{0}^{C}$. Multivariate backward stepwise step confirmed increased V_1_ with increasing BW (LRT = 8.11, *p* = 0.0044) and in males (V_1,males_ = 2.8 L, LRT = 6.17, *p* = 0.013), while CL was increased in males (CL_males_ = 0.23 L/day, LRT = 15.29, *p* = 9.2 × 10^−5^). In addition, UC was associated with increased $R_{0}^{C}$ compared to CD ($R_{0}^{C}$ = 5.8 nM, LRT = 5.71, *p* = 0.017).

### 3.3. Simulations

In simulations of 90% intervals of infliximab concentrations, total and unbound target levels and unbound/total target level ratios (R/R_T_) showed substantial differences between central and peripheral compartments and a large interindividual variability. Notably, the turnover of targets in peripheral compartment was slower than in the central compartment. As a result, before third and fourth infliximab infusions, while $R^{C}/R_{T}^{C}$ ratio re-increases above 30% in median, $R^{P}/R_{T}^{P}$ ratio remained at less than 3%, with a large interindividual variability (Figure 3). An infliximab serum concentration of 5 mg/L was associated with median $R^{C}/R_{T}^{C}$ and $R^{P}/R_{T}^{P}$ ratios of 26% and 1.2%, respectively. Since elimination half-life of infliximab is much higher than that of TNF-α, the amount maximum of total target is higher (10- to 100-fold) than baseline target amount.

## 4. Discussion

To our knowledge, this is the first study that investigated target-mediated elimination of infliximab in both central and peripheral compartments in inflammatory bowel diseases (IBD) using a double target-mediated drug disposition (TMDD) model. We showed that TNF-α turnover and its interactions with infliximab (infliximab-TNF complex formation and clearance) were substantially different between these two compartments.

Up to date, infliximab pharmacokinetics was studied using compartmental modeling in 36 publications, including 22 that used population two-compartment models, of which 15 in IBD patients [32,33]. The influence of target antigen on infliximab pharmacokinetics was suggested in several publications using covariates related to inflammation (CRP levels, fecal calprotectin, erythrocyte sedimentation rate) [16,32], and described using TMDD models in six studies, of which four were made on aggregated data [18,19,20,34], with two using population modeling [11,17].

As in our previous work, our estimates of baseline target levels (R_0_) in central (3.3 nM) and peripheral (0.46 nM) compartment were dramatically greater than circulating TNF-α levels (0.000038 nM [17]). As discussed in our previous publication [11], this suggests that our model measures antigen mass, both inside and outside the bloodstream. Nevertheless, the total antigen mass (i.e., central + peripheral) may be even higher: indeed, the present estimation is more than 10-fold that of our previous work (0.46 nM) and should be more accurate since our previous study was made using a one-compartment model. This suggests that most of TNF-α targeted by infliximab in the central compartment is not circulating in the bloodstream.

Moreover, our results suggest that infliximab-TNF-α interactions are very different in central and peripheral compartments. Notably, the elimination rate constant of infliximab-TNF-α complexes is 20-fold higher in central (0.17 day^−1^) than in peripheral (0.0079 day^−1^) compartment, suggesting a dramatically slower elimination of complexes and therefore a longer retention of infliximab in peripheral compartment. The value of central k_int_ is in agreement with median value reported for mAbs studied using TMDD modeling (0.13 day^−1^ [16]) but still inferior to the value estimated by Berends et al. (0.98 day^−1^ [17]). Indeed, this latter study quantified a rapid elimination of complexes due to blood circulating TNF-α, while the present, as well as previous ones [11,18,19,20], may have quantified a slower kinetics of complexes involving the whole antigen mass.

The difference in steady-state dissociation constant (K_SS_) estimates between central (15.4 nM) and peripheral (0.49 nM) compartments may be linked to target turnover and expression as well as to differences in infliximab-TNF affinities between central and peripheral compartment. In the central compartment, our K_SS_ estimate is similar to that reported by Berends et al. (14 nM [17]), while that of the peripheral compartment is similar to values linked to slow kinetics of complexes reported in previous publications on IBD patients from Kimura et al. (0.468 nM, [20]) and us (0.45 nM [11]). Of note, these latter values were dissociation constants (K_D_), but they should be similar to K_SS_ values because they were associated with low values of k_int_ [26,35]. The reasons of the 30-fold ratio between central and peripheral K_SS_ values are unclear. A possible reason might linked to the existence of two forms of TNF-α—i.e., homotrimers or monomers, trimeric TNF-α binding to receptors TNF receptors 1 and 2 and therefore the active form [36]. Trimers are known to monomerize for low concentrations [37], re-trimerization occurring if monomer concentrations are sufficient (>10 nM) [38]. Thus, low concentrations of circulating TNF-α are in favor of predominant monomer form in blood. Besides, concentrations of trimeric TNF-α are increased locally due to the presence of receptors [39] and due to anti-TNF, as infliximab, which stabilize trimeric TNF-α [40]. Trimeric TNF-α leads to the formation of infliximab-TNF complexes involving two or three molecules of each which are hypothesized to be more stable than simple 1:1 complexes [41]. Hence, the lower dissociation constant in peripheral compartment might be due to more stable complexes formed with trimeric than monomeric TNF-α.

It is generally considered that an efficient dosing strategy for mAbs should lead to sufficient target blockade, a variable which is often evaluated using the unbound/total target ratio (R/R_T_) [16,42]. Previous works made by Berends et al. [17] and us [11] showed that infliximab treatment was not associated with sustained TNF-α blockade, despite an absence of systematic loss of response. In our previous work [11], we had hypothesized a multi-scale turnover of TNF-α in IBD patients, including the existence of a deep compartment in which kinetics differed from that of the central compartment and which had still not been quantified. This compartment may be linked to TNF-α expressed on intestine inflammatory cells (monocytes, macrophages). The present work not only confirms that TNF-α is not durably occupied in the central compartment, but also suggests an almost full TNF-α blockade in peripheral compartment of most patients. This phenomenon suggests that sustained clinical response may be due to target blockade on inflammatory intestine cells.

Interestingly, our double TMDD model displays an apparent tri-phasic decay of infliximab concentrations in serum (Concentrations in central compartment, Figure 3), which suggests that infliximab pharmacokinetics in IBD patients is more complex than what was previously described. Indeed, estimates of two-compartment parameters varied markedly between studies, notably those of peripheral compartment. Notably, across the 15 publications of two-compartment kinetics of infliximab, there is a large disparity in estimates of intercompartment clearance (Q) [33]. Due to low values of Q (less than 0.1 L/day in average), one half of studies reported long distribution (T½-α > 3.5 days) and elimination (T½-β > 18 days) half-lives, while pharmacokinetic studies conducted with dense datasets reported high values of Q (3.7 L/day in average). Similarly, in the present work, values of Q were lower if no peripheral TMDD compartment was included (0.3 vs. 1.5 L/day, Table 2). The discrepancies in Q value estimates might therefore be related to the complex pharmacokinetic behavior of infliximab.

Conversely, publications that reported long T½-α were based on two-compartment models that may have not captured the actual distribution phase, but rather the ‘intermediate’ phase of the apparent tri-phasic elimination shape. This may be explained not only by an over-simplistic model, but also by data paucity [32] and differences in concentration measurement techniques [2]. The large differences in pharmacokinetic parameter values between studies may be an issue for concentration forecasting in therapeutic drug monitoring [43]. In that context, our TMDD model might overcome, at least in part, this issue.

Central baseline target amount was significantly higher in UC than in CD patients. This result is consistent with our previous publication in which both volume of distribution and clearance were higher in UC than in CD [44]. This difference is difficult to explain since no clear difference in TNF-α expression between CD and UC was reported [8,9]. This result may be linked to our cohort and may not be representative of all IBD patients treated with infliximab.

Our study has limitations. First, our model was developed using trough and peak concentration data only, as most of infliximab pharmacokinetic studies [32,33]. This prevented us from estimating TNF-α elimination rate constants in central and peripheral compartments, as well as interindividual variances of some parameters (*Q*, *K_SS_*, *k_int_*). Second, the elimination rate constant of TNF-α had to be fixed and assumed to be equal in both compartments. Of note, all previous infliximab TMDD models necessitated fixed values of dissociation constants (*K_D_* [11,18,19,20,34] or *K_SS_* [17]). Third, as in our previous work [11], we had to use a reference dataset for which we assumed no influence of antigen mass on infliximab pharmacokinetics, i.e., AS patients [21]. Even if no influence of antigen mass has been detected [27], such an influence cannot be definitely excluded. Nevertheless, the use of AS database as a reference was possible because all concentrations were measured using the same ELISA technique [24]. Fourth, both TNF-α were assumed as independent, which may be oversimplistic. Unfortunately, attempts to build a two-compartment model for TNF-α kinetics did not provide results that overcame our double TMDD model (Appendix A). Overall, due to these limitations, TMDD parameter estimates may have been biased and should therefore be considered with caution (Appendix A). Last but not least, no covariate linked to inflammation (CRP levels, erythrocyte sedimentation rates, fecal calprotectin [16,32]) or no clinical improvement endpoints were available in this database, which prevented us from interpreting our R_0_ estimates in both central and peripheral compartments and to investigate the link between TNF-α occupancy and clinical response.

## 5. Conclusions

This study provided the most complete description of infliximab pharmacokinetics in IBD patients. This model allowed us to take a step forward the understanding of the complexity of infliximab target-mediated pharmacokinetics which involves a multi-scale turnover of TNF-α. Nevertheless, further studies are still necessary to evaluate the relevance of our description of target occupancy kinetics. Our model should be applied to existing or upcoming data sets, which would ideally provide biological and/or clinical monitoring data, allowing the description of the relationship between target kinetics (unbound target or unbound/total target ratio) and monitoring data.

## Figures and Tables

**Figure 1 pharmaceutics-13-01821-f001:**
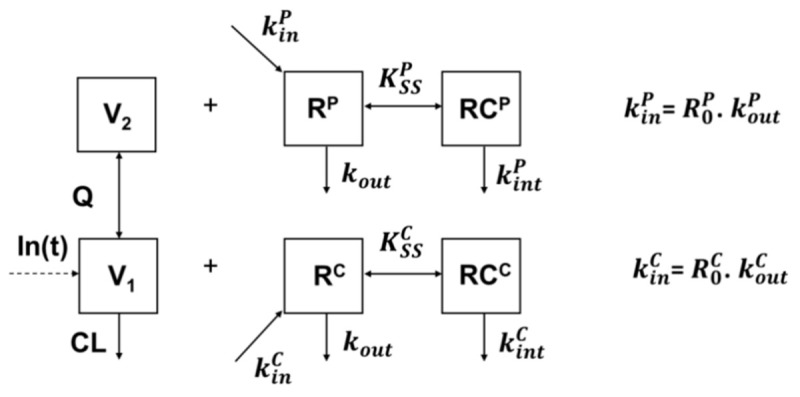
Target-mediated drug disposition (TMDD) with quasi-equilibrium (QSS) approximation. Base parameters are estimated for inflammatory bowel disease (IBD) and ankylosing spondylitis (AS), the latter being used as a reference, while TMDD parameters are estimated for central (below, “C” exponent) and peripheral (above, “P” exponent). *In(t)* is infliximab input function, V1 and V2 are central and peripheral volumes of distribution, respectively, CL and Q are endogenous and intercompartment clearances, respectively, R*_0_*, *k_in_*, and k_out_ are baseline TNF-α amount, zero-order unbound target production rate constant and first-order destruction rate constant, respectively, K_SS_ and k_int_ are steady-state dissociation constant and infliximab-TNF complex elimination rate constant, respectively.

**Figure 2 pharmaceutics-13-01821-f002:**
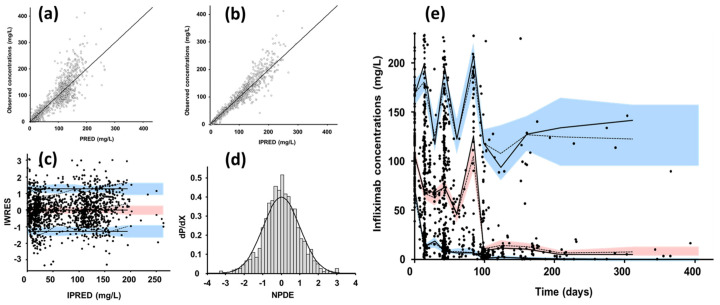
Diagnostic plots of the final pharmacokinetic TMDD quasi-steady-state (QSS) model. (**a**) Observed vs. population model fitted infliximab concentrations (PRED) and (**b**) observed vs. individual model-fitted infliximab concentrations, open circles are observed vs. fitted concentrations and line is first bisector; (**c**) individual weighted residuals (IWRES) vs. IPRED; black circles are IWRES vs. IPRED; (**d**) normalized prediction distribution error (NPDE) distribution vs. Gaussian probability density function; dashed line is theoretical Gaussian distribution; (**e**) visual predictive check; observed concentration (black circles), theoretical (dashed lines), and empirical (continuous lines) percentiles (from bottom to top: 10%, 50% and 100% percentiles) and prediction interval (from bottom to top: 10%, 50%, and 90% prediction intervals.

**Figure 3 pharmaceutics-13-01821-f003:**
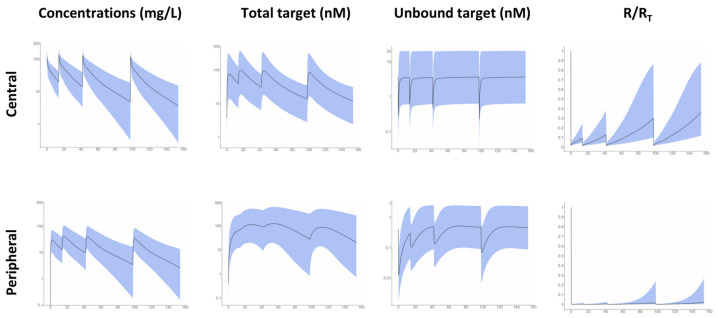
From left to right, 90% prediction intervals for infliximab concentrations, total target and unbound target levels, and unbound/total target ratio relative to central (**above**) and peripheral (**below**) compartments. Blue shaded are 90% prediction intervals and line is median profile.

**Table 1 pharmaceutics-13-01821-t001:** Summary of patient characteristics.

Characteristics	AS Cohort	IBD Cohort
Number of patients	25	133
Number of samples	488	845
Starting dose (mg)	400 (300–400)	300 (280–370)
Body weight (kg)	75 (65–85)	64 (56–72)
Age (years)	43 (35–52)	34 (25–41)
Sex (females/males)	6/19	53/80
Ankylosing spondylitis	25	─
Crohn’s disease	─	108
Ulcerative colitis	─	25

**Table 2 pharmaceutics-13-01821-t002:** Parameter estimates for different base models and final model.

Parameter	Unit	Model									
		Base 2 Compartments		Base TMDD Peripheral		Base TMDD Central		Base TMDD Central + Peripheral		Final TMDD Central + Peripheral	
		Estimate	RSE%	Estimate	RSE%	Estimate	RSE%	Estimate	RSE%	Estimate	RSE%
V_1_	L	3.0	2.4	2.8	2.6	2.9	2.5	2.8	2.5	2.6	3.3
CL	L.day^−1^	0.29	7.9	0.28	4.0	0.18	7.1	0.20	6.7	0.16	9.3
V_2_	L	2.2	6.9	1.8	8.3	1.9	9.8	1.9	11	1.9	8.8
Q	L.day^−1^	0.29	5.1	1.6	6.6	0.30	7.8	1.5	2.2	1.8	2.0
K^C^_SS_	nM	─	─	─	─	13.5	8.8	13.7	16	15.4	21
R^C^_0_	nM	─	─	─	─	7.2	21	2.6	21	3.3	28
k^C^_int_	day^−1^	─	─	─	─	0.089	11	0.13	17	0.17	11
K^P^_SS_	nM	─	─	0.49	5.5	─	─	0.45	5.1	0.49	11
R^P^_0_	nM	─	─	0.68	25	─	─	0.30	9.2	0.46	22
k^P^_int_	day^−1^	─	─	0.013	37	─	─	0.0050	33	0.0079	36
k_out_	day^−1^	─	─	20	(fixed)	20	(fixed)	20	(fixed)	20	(fixed)
BW_V_1_	─	─	─	─	─	─	─	─	─	0.33	35
SX_V_1_	─	─	─	─	─	─	─	─	─	0.13	40
SX_CL	─	─	─	─	─	─	─	─	─	0.36	26
UC_R^C^_0_	─	─	─	─	─	─	─	─	─	0.57	47
ω_V1_	─	0.28	6.8	0.29	6.8	0.26	7.0	0.29	6.8	0.27	7.0
ω_CL_	─	0.46	6.2	0.50	7.0	0.38	15	0.38	13.0	0.35	13
ω_V2_	─	0.62	9.8	0.32	24	0.71	12	0.36	24	0.39	25
ω_RC0_	─	─	─	─	─	0.98	15	1.0	14	1.0	15
ω_RP0_	─	─	─	0.31	6.8	─	─	1.2	20	1.1	16
σ_add_	mg/L	1.8	10	1.8	11	1.9	10	1.8	9.8	1.8	9.8
σ_prop_	─	0.20	3.1	0.20	3.4	0.20	3.2	0.20	3.0	0.20	3.0
−2LL	─	10,870.99	─	10,826.96	─	10,818.28	─	10,793.76	─	10,750.05	─
AIC	─	10,888.99	─	10,852.96	─	10,844.28	─	10,827.76	─	10,792.05	─

Legends. TMDD: target-mediated drug disposition; V_1_, V_2_: central and peripheral volumes of distribution; CL, Q: systemic and intercompartment clearances; K_SS_: steady-state dissociation constants relative to central and peripheral compartments; R_0_: baseline TNF-α amount relative to central and peripheral compartments; k_int_: infliximab-TNF-α complex elimination rate constant, “C” and “P” stand for central and peripheral compartments, respectively; k_out_: TNF-α elimination rate constant; *t*BW: body weight; SX: sex; UC: ulcerative colitis; −2LL: −2 ln-likelihood; AIC: Akaike’s information criterion.

## Data Availability

The data presented in this study are available on request from the corresponding author. The data are not publicly available because belong to the University hospital of Tours who has to authorize access to the database.

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
