# Peer review of "Infliximab Efficacy May Be Linked to Full TNF-α Blockade in Peripheral Compartment—A Double Central-Peripheral Target-Mediated Drug Disposition (TMDD) Model"

_pharmaceutics, 2021, doi:10.3390/pharmaceutics13111821_

Round 1
Reviewer 1 Report
In the present investigation, authors have used a double central-peripheral target-mediated drug deposition model to find the relation between Infliximab efficiency and TNF-a blockage in the chronic inflammatory disease patients. The authors found different 20 fold higher in central compartment than the peripheral compartment and indicate longitivtiy in peripheral compartment. The model given by the authors may be very useful in monitoring the clinical data and also to understand the impact of drugs on the biological systems. The hypothesis is reasonably good. The manuscript is well executed and nicely written. The manuscript may be accepted for the publication in the present form.
Author Response
Thank you for this nice review.
Reviewer 2 Report
This paper is a very interesting study on the multi-scale target-mediated pharmacokinetics of infliximab. These studies allow new understandings about the pharmacokinetic behavior of biologics for different diseases. However, some clarifications to understand and improve the document:
1. What are the inclusion and exclusion criteria for the patient cohort. I don't see it explicitly described in the document
2. Regarding the results obtained by ELISA, a graph is not shown showing the range and concentration levels of INFLIXIMAB.
3. Describe in the paper why 1,333 concentrations of infliximab were obtained
4. This behavior of the pharmacokinetic model could be extrapolated to other biologics other than Infliximab.
5. For the case of a study comparing MDD of infliximab and the favorable clinical outcome of the patients, could a similar pharmacokinetic model be designed?
Author Response
This paper is a very interesting study on the multi-scale target-mediated pharmacokinetics of infliximab. These studies allow new understandings about the pharmacokinetic behavior of biologics for different diseases. However, some clarifications to understand and improve the document:
Thank you for this nice review. Please find below answers to reviewer's concerns :
- What are the inclusion and exclusion criteria for the patient cohort. I don't see it explicitly described in the document
Answer : We agree that inclusion/exclusion criteria for SPAXIM database (ankylosing spondylitis patients, N=26) were not explicitly given because were given in our previous publication (Ternant et al. Br J Clin Pharmacol, 2012). Please find these precised here: « To be eligible, patients had to be adult (18 years of age or older), with AS according to modified New York criteria [19], with an indication for anti-TNF-α treatment and no contraindication for either infliximab or methotrexate. Treatment with NSAIDS before or during the study was allowed. Patients were not included in the study if they were pregnant or subject for pregnancy, breast feeding, addicted or intoxicated to alcohol or drugs within one year prior to the inclusion, participating to another clinical study; if they had or were already treated with infliximab or methotrexate; if they had white blood cells < 2,000 mm3, hemoglobin < 9 g/dL or platelets < 105 mm3, an active malignancy within the 5 years prior to the inclusion, severe or persistent infections requiring hospitalization or intravenous antibiotic treatment within 30 days prior to inclusion, severe chronic disease (B or C hepatitis, HIV, active or latent tuberculosis, demyelinating disease, or renal, hepatic, hematological, endocrine, pulmonary, cardiac, neurological or brain evolutive disease), or a planned surgical intervention during the study period. ». With reviewer’s and editor’s authorization, we would like not to repeat these criteria which we do not believe are useful here. However, we precised that « Inclusion and exclusion criteria were previously described ». (section 2.1.). The only change of this study was removal of a patient with antidrug antibodies (ADA). This is given in text (end of first pragraph of section 2.1.). As for retrospective IBD cohort (N=133), we precise in 2nd paragraph of section 2.1. inclusion criteria, but not clearly exclusion criteria. Patients were excluded if less than 3 concentration values were available (7 excluded patients of our database between 2006 and 2012), if no peak concentrations were available (2 more patients excluded), if ADA were dectected at first, second of third cycles (3 more patients excluded). We now give these information in the second paragraph of section 2.1. We now give these information in the second paragraph of section 2.1. Of note, 5 patients were excluded because less than 3 concentration available and no peak concentration available).
- Regarding the results obtained by ELISA, a graph is not shown showing the range and concentration levels of INFLIXIMAB.
Answer : We understand that the rewiewer would like to see a general chart showing all infliximab concentrations. We therefore added a spaghetti plot showing infliximab concentrations in all patients in time (supplemental data part 1, please find this plot in attached document and/or supplemental data). For a better representation, IBD and AS databases were splitted.
Figure S1. Spaghetti plots representing observed infliximab concentrations in time for inflammatory bowel disease patients (left) and ankylosing spondylitis patients (right).
- Describe in the paper why 1,333 concentrations of infliximab were obtained
Answer: The number of samples per patient was superior in AS database (488 observations in total, 19.5 observations per patient in average) than in IBD database (845 observation in total, 6.3 observations per patient in average), with total number of 1333 concentrations. We have now added in table 1 total numbers of observations in both databases.
- This behavior of the pharmacokinetic model could be extrapolated to other biologics other than Infliximab.
Answer: We agree with the reviewer, this analysis could be made in all biologics provided that (i) the availability of sufficiently dense concentration-time data and (ii) detectable nonlinear target-mediated pharmacokinetics. This was already done for several mAbs, including rituximab (anti-CD20), cetuximab (anti-EGFR), bevacizumab (anti-VEGF), trastuzumab (anti-HER2), canakinumab (anti-IL-1beta), omalizumab (anti-IgE).
- For the case of a study comparing MDD of infliximab and the favorable clinical outcome of the patients, could a similar pharmacokinetic model be designed?
Answer: Usually, concentration-response relationship of mAbs in general and infliximab in particular is described by linking concentrations and (biological and/or clinical) response. Our belief is that a better description would be made by using a TMDD model to estimate target kinetics and to describe the relationship between target kinetics (unbound target or unbound/total target ratio) and biological/clinical outcome. This was the sense of our last discussion sentense « to investigate the link between TNF-alpha occupancy and clinical response ». an estimate of unbound target (or unbound/total target ratio) at a given time or at steady-state could be used as a covariate for logistic regression (response vs. nonresponse). We therefore added a statement in the last sentense of conclusion « … alowwing the description of the relationship between target kinetics (unbound target or unbound/total target ratio) and monitoring data. ».

Reviewer 3 Report
The manuscript is interesting however should emphasize why they chose the covariates. How did they improve the two-compartment model. Did Authors consider to use classic two-compartment model or non-compartmental analysis ?
Author Response
Thank you for this nice review.
- We agree that covariate search strategy should be precised here. As infliximab was already described in previous reports, we limited covariate search to those which were already found in previous analysis. Indeed, because of both model complexity and relatively large number of patients (122) and of observations (1333), runs lasted up to 9 hours, we had to limit the number of runs as low as possible. We now added in 2.2.2 section (influence of covariates) that "Covariates that were considered in this analysis were those which were found significantly associated with pharmacokinetic parameter in previous studies involving these databases, i.e. body weight (BW), sex (SX) and underling disease".
- In section 3.2 we gave the likelihood ratio test results of each covariate association with PK parameters and p-values: increasing central volume was significatnly associated with increased body weight and was higher in males, while systemic clearance was higher in males. In addition, ulcerative colitis patients presented slightly but significantly higher baseline target level than Crohn's disease patients. "Global" influence of covariates can be found in table two by comparing -2 likelihoods of "base TMDD central+peripheral" and "final TMDD central+peripheral" models, for which LRT was 43.71.
- We indeed tested a base 2 compartment model, which results are available in table 2 (left). as well as "simple" TMDD models with interactions in central or peripheral compartments. The double TMDD model led to a significantly better description than all these models, with respective LRT of 77.23, 33.2 and 24.52. This is confirmed with inspection of Akaike's information criterion (AIC) which is lower for double TMDD model than all other models.
- Noncompartmental analysis would have been suitable for SPAXIM database (ankylosing patients, N=25) because several observations were available to describe terminal elimination phase. However, this approch would not have been possible in IBD patient database, because only trough and peak samples during infliximab treatment were drawn. If rich IBD patient database had been available, were are not sure to have bee able to assess elimination rate constant (lambda-z), because no target level was measured. The superiority of TMDD modeling approach is to be able to estimate target level kinetics associated with nonlinear PK of infliximab.
Reviewer 4 Report
The manuscript is well written and scientifically sound. The results have been placed in context of previous publications and the discussions are clear and logically explained.
Author Response
Thank you for this nice review.